# Compiler Toolchains for Deep Learning Workloads on Embedded Platforms

## Abstract

As the usage of deep learning becomes increasingly popular in mobile and embedded solutions, it is necessary to convert the framework-specific network representations into executable code for these embedded platforms.

This paper starts with a survey and benchmark of the available open source deep learning compiler toolchains, which focuses on the capabilities and performance of the toolchains in regard to targeting embedded microcontrollers that are combined with a dedicated accelerator in a heterogeneous fashion.

The second part focuses on the implementation and evaluation of a compilation flow that targets such a solution and reuses one of the existing toolchains to demonstrate the necessary steps for hardware developers to build a software flow for their product.

*CCS Concepts:* • **Computing methodologies → Artificial intelligence**.

*Keywords:* deep learning, embedded, deep learning compiler

**ACM Reference Format:**
Anonymous Author(s). 2020. Compiler Toolchains for Deep Learning Workloads on Embedded Platforms. In *Proceedings of Burlingame '21: First International Research Symposium on Tiny Machine Learning (tinyML) (Burlingame '21)*. ACM, New York, NY, USA, 8 pages. https://doi.org/10.1145/nnnnnnn.nnnnnnn

## 1 Introduction

As AI is moving to the edge, the limitations of the popular deep learning frameworks in regard to embedded platforms become apparent. These frameworks are mostly designed for the server and workstation use and incorporate many features that are not relevant for the inference on low power devices. This prevents them from running on microcontrollers and other embedded devices.

Due to this, the deployment of deep learning models on embedded devices typically relies on the manual implementation of the previously trained network. The developers have to implement the required layer types, preferably using the vendor-optimized math kernel libraries of the platform. This process is labour-intense, error prone and can easily

*Burlingame '21, March 22, 2021, Burlingame, CA*
© 2020 Association for Computing Machinery.
ACM ISBN 978-x-xxxx-xxxx-x/YY/MM...$15.00
https://doi.org/10.1145/nnnnnnn.nnnnnnn

result in inferior performance due to missing optimizations. In addition, the support of new platforms can result an extensive effort as the function kernels might need to be reimplemented for them.

The necessary effort increases even further if custom dedicated deep learning accelerators are employed. Due to their use of domain-specific instruction sets, which often utilize longer pipelines that cover common neural network subpatterns, these accelerators cannot be easily targeted by standard compiler toolchains. Compiler backends for such devices can be difficult to implement due to the coarse-granularity of the accelerator instruction set architectures.

A solution to automate these tasks are deep learning compilers. These toolchains operate similar to standard compilers, but introduce a number of important peculiarities: Instead of handwritten source code, deep learning compilers process serialized trained neural network descriptions. Additionally, they are able to automatically employ the optimized math kernel libraries or alternative optimizations of the function kernels. A benefit of employing domain-specific compilers is the option to introduce additional optimizations that target deep learning models. Typical optimizations are layer fusion or general network graph optimizations and quantization schemes. Lastly, these toolchains are able to target heterogeneous platforms and dedicated accelerators by employing runtimes on the target devices. These runtime take care of the scheduling and additional support operations that are necessary as well as the deserialization of the compiled networks.

This paper will start with an overview of the available optimizations and math kernel libraries for deep learning workloads on embedded platforms. These should be incorporated by the compiler toolchains. The following sections will cover a survey of the compiler features and the achieved performance on different embedded and low-power platforms, while the last part of the paper contains the implementation of a compilation flow for a new custom target.

## 2 Related Work

Two recent studies focus on the employed techniques and the use with with FPGA platforms [49], as well as an indepth overview over the different approaches for common problems of the available deep learning compilers [30]. In contrast to these publications, this work focuses on embedded platforms. In addition to the survey and benchmark a compilation toolchain for a new device has been implemented. This was done to demonstrate the steps that are currently required to support custom accelerators with a software stack.

## 2.1 Deep Learning Optimizations

Deep Learning toolchains can employ a multitude of domain-specific optimizations in addition to standard compiler optimizations. These include weight pruning, which cuts redundant and unnecessary weights from the networks to reduce its size and - depending on the implementation - the compute workload. While a wide range of pruning algorithms exist, none of them are currently employed by deep learning compilers [4, 29, 41, 47].

In contrast, quantization schemes are employed by most deep learning compilers. This optimization converts the floating point representations for weights, intermediate results and outputs into fixed-point formats, while mostly keeping the accuracy of the original network. This enables the network to take up less storage and reduces its bandwidth requirements during the execution as well as the computational intensity, if optimized function kernels for the quantized representations exist [23, 25, 32, 42, 48].

Additional strategies include optimizations on the neural network graph like layer fusion, dead node elimination and others [27, 43].

## 2.2 Math Kernel Libraries

Math kernel libraries are a way to provide developers with optimized function kernels for common operations, increasing the efficiency of software solutions that employ them, while decreasing the redundancy of implementations. They are typically platform-specific and provided by the device-vendors[1]. Table 1 presents an overview of the supported platforms per library. These libraries differ largely in their implementation strategies and offered functionality. While all libraries provide function kernels, their implementations follow different approaches: ARM CMSIS-NN [26] and Nvidia cuDNN [11] mostly resort to a low number of kernels that deliver consistent performance across the majority of the configuration space. Solutions like Intel's oneDNN[2] [14] library implement most operations with multiple different strategies, based on the available instruction set extensions, data types and the layer configuration. oneDNN then selects the best suited function kernel at runtime[3]. While this strategy can be able to achieve better performance in certain edge cases, it requires much more maintenance and implementation effort compared to the more general function kernels.

## 2.3 Deep Learning Accelerators

In recent years plenty of deep learning accelerators emerged in commercial and research applications. Commercial accelerators for embedded platforms include the NVDLA from Nvidia [36], Google's EdgeTPU [5] as well as ARM's Ethos-U

---

[1]Efforts for device-independent solutions exist as well, but have not found the same rate of adaption, e.g. XNNPack [19]

[2]previously known as MKL-DNN

[3]only for its x86_64 CPU backend

[4]only supports GAP processors[17] from Greenwave technologies

[5]experimental, JIT approach implemented by Fujitsu[31]

[6]Intel GPUs only, no JIT solution, static instead

[7]OpenPower & IBM Power, both experimental, no JIT

[8]also experimental, no JIT

**Table 1.** Overview over the available math kernel libraries and their supported platforms

| | Cortex-M | RISC-V | ARM64 | x86_64 | CUDA | OpenCL | PowerPC | Others |
|---|---|---|---|---|---|---|---|---|
| CMSIS-NN[26] | ✓ | ✗ | ✗ | ✗ | ✗ | ✗ | ✗ | ✗ |
| PULP-NN[18] | ✗ | ✓[4] | ✗ | ✗ | ✗ | ✗ | ✗ | ✗ |
| cuDNN[11] | ✗ | ✗ | ✗ | ✗ | ✓ | ✗ | ✗ | ✗ |
| MIOpen[24] | ✗ | ✗ | ✗ | ✗ | ✗ | ✓ | ✗ | ✗ |
| oneDNN[14] | ✗ | ✗ | ✓[5] | ✓ | ✗ | ✓[6] | ✓[7] | IBMz[8] |
| XNNPack[19] | ✗ | ✗ | ✓ | ✓ | ✗ | ✗ | ✗ | x86, ARMv7 +NEON |

NPU [2]. Most of these solutions employ custom compilation toolchains that are limited to support of only one deep learning framework for its input formats.

Research platforms include the Eyeriss accelerators (v1, v2) [8, 9], VTA [35] as well as a many FPGA-based solutions [40]. These typically do not focus on the software toolchain and explore novel architecture approaches instead.

FPGA deep learning high-level synthesis frameworks are also mentioned here as they represent a mixture of deep learning compilers and accelerators. These frameworks synthesize FPGA overlays from the deep learning models that are able to execute them [46].

## 3 Survey & Benchmark

The survey section of the paper covers open source projects that are still in development. Its focus lies on the inference of deep learning models on embedded hardware. The support for the training step will not be evaluated as it is uncommon to execute it on the embedded device itself.

The evaluated deep learning compilers are TensorFlow Lite (TFLite) [44], TensorFlow XLA (TF XLA) [45], Glow [39], TVM [7], ONNC [33] and nGraph [15], which has been tested as part of Intel's openVINO toolkit.

### 3.1 Common Strategies

All evaluated toolchains follow the typical compiler structure. The frontend converts the serialized pretrained models into its high-level intermediate representation (IR). Most toolchains utilize a two-level IR: The high-level IR is a graph-level representation of the compiled model and the low-level IR describes the operations on the tensor level. The graph-level IR is typically used for target-independent optimizations and operator fusion. The tensor-level IR is used by the backend to optimize the individual layers for the designated target device.

One exception is TFLite, which does not perform target-dependent optimizations at the compilation stage. Instead, its compiler (TFLite converter) generates a graph-level representation, that does not contain execution details, as the device-specific function kernels are part of the runtime. This

allows for a better portability of the compiler output across devices, but prevents more target-specific optimizations at the offline compilation stage.

The majority of the evaluated toolchains employs a runtime[9], which needs to be present on the target device to execute the compiled neural network. The functionality of the runtime differs between projects. All runtimes provide support infrastructure to unpack the compiled neural networks and an API for the integration into the user program. Solutions like TFLite and ONNC deliver the operation execution strategies for the platform as part of the runtime. Glow and TVM utilize the runtime for heterogeneous targets and in addition TVM requires the runtime for data collection during the auto-tuning step. TVM delivers function kernels that have been generated by its Auto-tuner in a separate file alongside the model description.

The main difference between the evaluated projects is the provisioning of the function kernels. Most solutions utilize handcrafted implementations that integrate math kernel libraries. This requires maintenance and updating of implementations for each layer type across all supported platforms[10]. To circumvent these limitations, TVM employs an Auto-Tuning solutions which tries to find the best suited function kernels by using an AI-guided flow that incorporates measurements and estimations of execution times on the real target. Glow bypasses all of these problems by reusing the same generalized function kernels across all targets, where its target-dependend optimizations are only applied by the standard compiler backend for general purpose targets.

## 3.2 User Accessibility

The user accessibility mostly depends on the user interface of the offline compiler stage, the integration of the compiler output into the target application and the supported input formats.

For the supported frameworks and formats, ONNX[37] is the most important as it is an industry standard and converters from most frameworks exist. See table 2 for an overview of the supported formats and frameworks of each compiler toolchain.

All compiler stages either support a command-line interface, like traditional compiler toolchains, or the use through a Python API, which allows for the integration in the original training script of the deep learning model. One exception is Intel's openVINO that provides an additional graphical user interface through a web-interface [21]. This enables more direct feedback to the developer on the impact of different optimizations on the overall model performance.

For the integration in the user application, all toolchains provide a C or C++ API. TVM and TFLite provide an additional Python interface through their standard runtimes[11].

**Table 2.** Overview of the supported deep learning framework formats.

| DL Framework | TVM | TF Lite | TF XLA | Glow | ONNC | openVINO |
|---|---|---|---|---|---|---|
| ONNX[37] | ✓ | ✗ | ✗ | ✓ | ✓ | ✓ |
| TensorFlow[1] | ✓ | ✓ | ✓ | ✗ | ✗ | ✓ |
| TensorFlow Lite flatbuffer | ✓ | ✓ | ✗ | ✓ | ✗ | ✗ |
| PyTorch[38] | ✓ | ✗ | ✗ | ✗ | ✗ | ✗ |
| MXNet[6] | ✓ | ✗ | ✗ | ✗ | ✗ | ✓ |
| Caffe[34] | ✓ | ✗ | ✗ | ✓ | ✗ | ✓ |
| Keras[12] | ✓ | ✓ | ✓ | ✗ | ✗ | ✗ |

## 3.3 Supported Platforms & Extensibility

The range of supported platforms varies between the evaluated toolchains. In addition, the level of optimization for the supported platforms fluctuates widely. One such example is TFLite's support of the x86_64 platform: While its runtime can be compiled for it, the function kernels are not optimized, resulting in worse performance compared to other platforms or compilers.

For TF XLA no conclusive information about its support for different architectures could be found, as the official documentation and publications contradict each other [20, 28]. The support for bare-metal use cases and embedded processors is much less common, as only TFLite, ONNC and Glow are able to target Cortex-M CPUs. For an overview of the supported platforms see table 3.

While all toolchains include some kind of support for heterogeneous platforms, the implementation differs between them. The most complete solution has been implemented by TVM in its Bring-Your-Own-Codegen (BYOC) flow [10], that allows developers to target new libraries and accelerators from TVM's high-level IR. It does not only provide an API to include new backends, it also supports the developer by implementing solutions for common problems, like the CPU fallback for unsupported operations, an infrastructure for custom operator fusion rules and the option to change the data layout of tensors. Most other toolchains only supply a simple API and require the developer to reimplement many of these tasks.

A stark contrast to TVM is TFLite. Its compilation stage does not provide an interface for the inclusion of additional target-specific tasks and optimizations. New platforms are targeted by porting the runtime to them and deploying optimized function kernels with it. As this flow only allows the targeting of general purpose hardware, its support for the currently available accelerators has been realized by additional tools. These modify the flatbuffer file, which has been generated by the offline compilation stage, before it can be executed on the accelerator platform inside a modified runtime [3, 13].

---

[9]The only exception that never uses a runtime is TensorFlow XLA, Glow's AOT flow for CPUs does not require a runtime on the target

[10]TFLite, ONNC and others employ this strategy

[11]TFLite provides an additional runtime for microcontrollers, which does not come with a Python API

This approach breaks the portability of TFLite's flatbuffer files and requires additional work to keep these tools compatible with the current scheme of the TFLite flatbuffer files. Some compiler toolchains like Glow, which reuse LLVM's backends[12] can easily target new architectures, if an LLVM already exists. In that case Glow's AOT compilation mode can be reused, if other targets need to be supported, a separate runtime needs to be used and the AOT flow can no longer be utilized.

**Table 3.** Overview of the supported hardware by compiler toolchain.

| Hardware | TVM | TF Lite | TF XLA | Glow | ONNC | openVINO |
|----------|-----|---------|--------|------|------|----------|
| x86_64 | ✓ | ✓ | ✓ | ✓ | ✓ | ✓ |
| Cortex-A | ✓ | ✓ | ?[13] | ✓ | ✓ | ✗ |
| Cortex-M | ✗ | ✓ | ?[13] | ✓ | ✓ | ✗ |
| GPU (CUDA) | ✓ | ✗ | ✓ | ✗ | ✗ | ✗ |
| GPU (OpenCL) | ✓ | ✓ | ✓ | ✓ | ✗ | ✓ |
| Bare-Metal | ✗ | ✓ | ?[13] | ✓ | ✓ | ✗ |
| DLA | ✓ | ✓ | ✓ | ✓ | ✓ | ✓ |

### 3.4 Features

For the embedded use case, the AOT compilation and quantization support are the most important compiler features. For the optimal execution on the target devices, the toolchains should be able to incorporate math kernel libraries or auto-tuning to provide optimal function kernels for the execution. Features like support for the backpropagation and training steps are not as important for embedded devices (yet) and are only supported by TF XLA, Glow and openVINO as they mostly target HPC and cloud applications.

For the optimization of the layer execution TFLite, ONNC and openVINO rely on math kernel libraries, while Glow utilizes the same set of generalized function kernels across all targets, only utilizing the LLVM backend optimization steps. TVM is the only evaluated toolchain that employs an auto-tuning process to find good performing function kernels automatically.

All toolchains except TVM only implement a limited number of static quantization schemes with fixed bit-widths for intermediates and weights. This is a limitation for the targeting of devices that employ different schemes in their hardware implementation. TVM has implemented a more flexible quantization system, that offers different sizes and requantization strategies. However, it is more difficult to configure it correctly and to find a suitable accuracy compared to the other solutions.

---

[12]for its AOT flow
[13]inconclusive data

### 3.5 Performance

The performance was tested on an ARM Cortex-M55 fast model[14], an Cortex-A72[15] and a Intel Celeron J1900[16]. All of these platforms provide a SIMD vector extension and have been tested with the same simple MNIST test network, consisting of convolutional, fully connected, ReLU and maximum pooling layers. The batch size has been set to one and the final activation function has been removed after training, as it is common in embedded applications.

The Cortex-M55 could only be targeted by TFLite[17] and

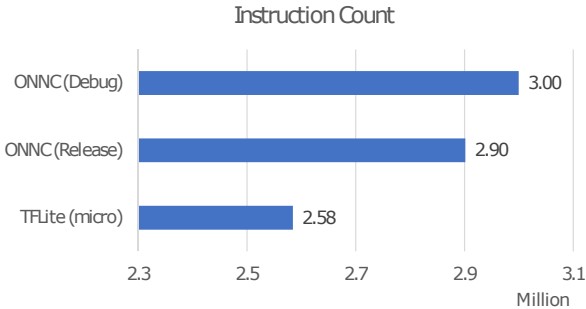

**Figure 1.** The instruction counts for a single inference that have been estimated by the Cortex-M55 fast model.

ONNC[18]. While Glow is able to target Cortex-M CPUs, it was not able to support the novel instruction set of the platform (ARMv8.1M). The testing showed that TFLite required less instructions to complete an inference run (2.6 M instead of 3 M instructions, see figure 1), while ONNC allocated significantly less memory (1.6 KiB instead of 3 KiB. See figure 2 for details).

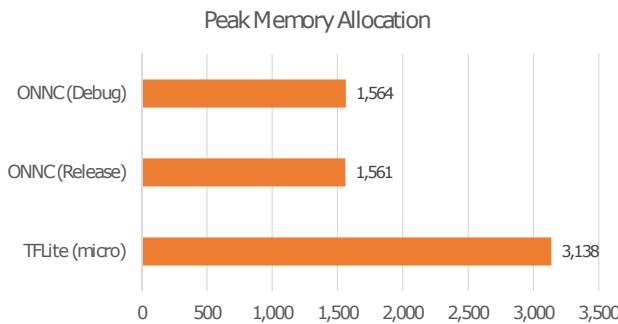

**Figure 2.** The peak allocations of RAM and ROM for the available toolchains during a single inference (batch size = 1).

The next test platform was the Cortex-A72. ONNC could not

---

[14]as no hardware was available at the time of testing
[15]using a Raspberry Pi 4 with 4 GB of system memory
[16]using 8 GB of DDR3 system memory
[17]using its micro-runtime
[18]using a special Cortex-M version of the toolchain

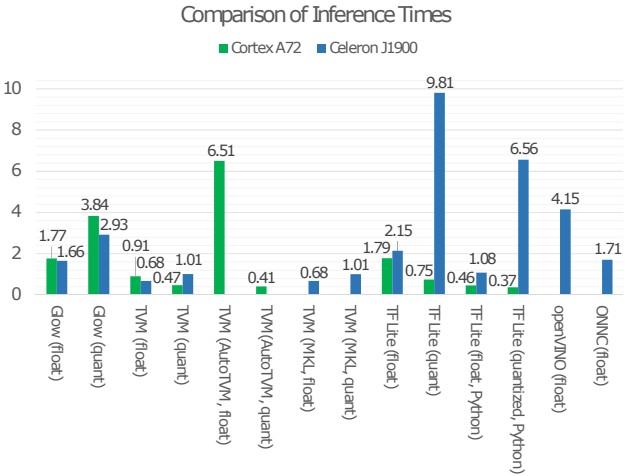

**Figure 3.** Comparison of the inference times on the ARM Cortex-A72 and Intel Celeron J1900 platforms.

be compiled for it, as it relied on Intel MKL for its function kernels[19]. Instead Glow, TVM and TFLite have been tested in addition to the standard TensorFlow Python runtime. An overview of the measured inference times can be seen at figure 3.

The quantized TFLite version achieved the fastest inference speed with 0.37 ms, followed by the quantized and auto-tuned TVM program, using the C API of its runtime (0.41 ms). Glows fastest output achieved a mean inference time of 1.77 ms by using floating-point representations. Besides Glow, all compilers achieved faster inference times using quantized networks - suggesting that they employ optimized function kernels for quantized operations, while Glow uses its generalized standard kernels and wraps them into additional quantization and dequantization steps, which causes additional overhead. The worst result by a compiler was achieved by TVM, for its floating-point auto-tuned solution, which tried to apply x86 function templates to the ARM platform[20]. However, the slowest result of 6.51 ms was still significantly faster than the use of the network in combination with the standard TensorFlow runtime - requiring 104.64 ms for a single inference run. This makes the slowest and incorrectly optimized compiled version 16 times faster, while the fastest compiled version achieved a speedup of 282.6 times.

The Intel Celeron CPU allowed for the additional testing of nGraph[21] and ONNC's standard flow[22]. See figure 3 for the inference time results of the platform. In comparison to the Cortex-A results the ranking of the toolchains by their

---

[19]While the CMake script for the standard runtime contained a parameter to disable the MKL integration, it could not be build when it was selected

[20]it could not be determined, if it was caused by user error or by TVM, but it reoccured over multiple tries and did not occur with the quantized version

[21]as part of openVINO

[22]besides its Cortex-M version

inference time changed, suggesting different levels of optimizations across the supported target devices for some deep learning toolchains. In addition, TVM was tested with the Intel MKL BYOC-based backend instead of its auto-tuning flow. This backend is not optimized for performance as it is a demo for the BYOC functionality and was used to estimate the overhead which results from it. For the Celeron J1900, the floating-point versions of the compiled networks achieved faster inference speeds across all toolchains. This suggests either a lack of optimized kernels or a better implementation of the floating-point components of the hardware itself. The fastest results have been achieved by TVM with 0.68 ms (FP) and 1.01 ms (quantized). TVM did not show a significant difference between the standard and the BYOC flow results, which implies that the overhead of the BYOC flow is neglectable. The next best results were achieved by TFLite's floating point network (1.08 ms, using the Python API), Glow (also floating point, 1.66 ms) and ONNC (1.71 ms). openVINO's compiled program did require 4.15 ms for a single inference, which made it the slowest floating point version out of the tested compiled networks. It was not able to quantize the network, as that is only supported on newer CPU platforms. Only TFLite's quantized networks took more time than openVINO to complete their inference run.

In addition to the inference times, the peak memory allocations have been measured. The measured results varied by two orders of magnitude between the toolchains. Glow's compiled networks required 10.11 MiB at peak, followed by TVM with 21 MiB to 26 MiB. As the higher allocations have been measured for the MKL-BYOC variant, it suggests, that the BYOC flow requires some memory overhead compared to the standard flow during the execution. TFLite required 13.6 MiB for a quantized version of the network utilizing only its C-API, which took significantly longer than the other results for a inference. The same configuration, but with a floating point version of the network allocated 237.65 MiB which is more than the expected increase by four times[23]. ONNC could only be tested with a floating point network as its open source standard branch does not support quantization. Its peak memory allocation of 51.18 MiB is more in line with the expected memory allocation. openVINO's implementation allocated 489.28 MiB of memory during the inference, only TFLite's Python runtime used more memory with 895.76 MiB (quantized) or 1,248.30 MiB (floating point). The prediction accuracy for the compiled networks stayed mostly the same, even for the quantized variants, with the exception of TVM. It only reached an accuracy of around 50 %, which might have been user error due to its configurable quantization scheme.

## 4 Implementation

For the implementation, an abstract accelerator has been defined and an instruction set simulator was implemented. The simulator was verified with TensorFlow and is able to estimate the hardware utilization and cycle count for input and output parallel execution strategies.

The simulated accelerator uses an instruction set that is similar in its capabilities to other solutions like Nvidia's NVDLA

---

[23]as 8-bit integers are 4 times smaller than 32-bit floating point values

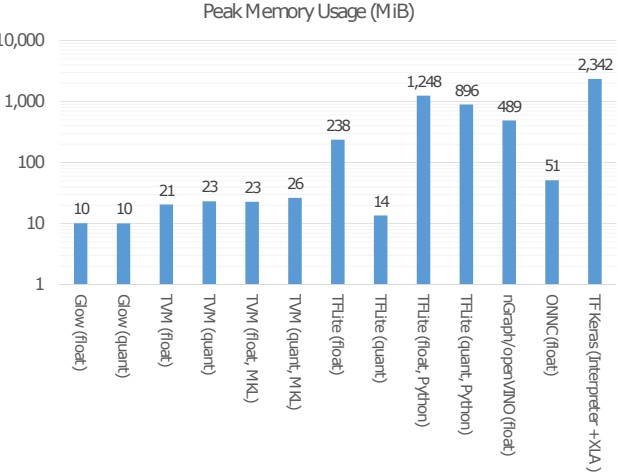

**Figure 4.** Comparison of the peak memory allocation on the Intel Celeron J1900 platform.

[36]. It only supports signed integer formats for operations and the majority of them are limited to eight bit.

For the software flow TVM was used due to its BYOC functionality. This flow starts with the definition of annotation rules for supported nodes and patterns in TVM's graph-level IR[24]. These are then merged into subgraphs, which will be executed by the accelerator. TVM manages the execution of unsupported operations on the CPU as well as the invoking of the subgraphs from the standard runtime. After the annotation, the network graph is partitioned to separate the supported sections of the network into subgraphs. These subgraphs are then passed on to the custom code generation, where they are converted into a JSON format for better portability across different instruction set variants. The final generation of the accelerator command stream happens at runtime before the initial inference. This allows to target different ISA variants with different memory sizes with a single serialized file. A custom runtime component executes the code generation and passes back a run function for each subgraph to the standard TVM graph runtime.

Besides the quantization and data layout transformation functionality, which was provided by TVM, the memory planning for DMA operations between system and accelerator memory, the assembly generation, the configuration register file management and tiling for larger tensor sizes needed to be implemented by the custom runtime component. The tiling was implemented by primarily splitting the workload along the output channel dimension.

## 5 Evaluation

The correct functionality was initially tested with neural networks that only contained supported operations as single nodes and patterns. Additional testing with the MNIST network from the performance benchmark revealed that the current TVM quantization flow inserts additional meta nodes into the graph. These nodes prevent the merging of

---
[24]called Relay

multiple compute layers into a single subgraph. Due to this, the network was split into three subgraphs, which requires the system to move the intermediate results back to the shared system memory between the subgraph executions. This results in reduced performance and efficiency due to unnecessary bus transactions. Otherwise, the custom backend worked as intended and generated the expected inference results.

For a more realistic test case Google's MobileNetV1 [22] with the ImageNet dataset [16] has been used. The additional batch normalization layers prevented the use of the larger convolutional pipelines, as they are located between the convolutional and activation function nodes. This adds additional bus transactions between accelerator memory and compute units, which reduces the efficiency of real hardware. The network was evaluated with three iterations of the accelerator and its toolchain:

- Without support for depthwise convolutions:
  These layers are executed by the standard runtime on the CPU. This results in a lower share of the network to be accelerated by the target device. The estimated cycle counts can be seen in figure 5a.
- Software-only Implementation:
  The depthwise convolutions are mapped to a the standard convolution instruction of the DLA. The ISA has not been changed. This allows for a larger share of the network to be accelerated, but the depthwise layers are constraint to a small utilization of the processing elements.
- Full support:
  The ISA has been extended to provide a dedicated instruction for deptwhise convolutional layers. This instruction allows for a higher utilization, resulting in shorter execution times.

This was done to evaluate the flexibility of the coarse-grained ISA for the support of new layer types, as the deep learning field develops new layer types at a very rapid pace, which makes it difficult to develop accelerators that can stay up-to-date without changing the hardware architecture. A solution would be to update the software flow, to enable it to map new layers to existing instructions. In the case of deptwhise convolutions, this approach was represented by the second test scenario. While the implementation was possible, it resulted in drastically increased cycle counts if compared to a native solution (compare figure 5b and 5c). This was caused by the low utilization of the PEs as only one filter could be processed at a time.

Additionally, these scenarios were tested with different sizes of the accelerators exclusive memory and the vector ALU as well as input and output parallel mode for cycle and utilization estimations to evaluate the impact of operation tiling on the overall performance. The evaluated memory sizes were 512 KiB and 256 MiB. The last configuration does not require any tiling to take place, while the first is the smallest size which is supported by the implementing tiling methods[25]. As shown in figure 5a, the output parallel hardware achieved

---
[25]For convolutional layers only a split along the output channel dimension was implemented, as the splitting along the rows and columns requires extensive effort to implement and validate all edge cases that can occur

lower cycle counts due to its higher utilization in the early layers of the network. However, this changed as soon as tiling was required due to the limited amount of device memory. As the tiling splits the workload along the channel dimension of the output tensor, the parallelization opportunity for this architecture shrinks. This results in a lower utilization, and a faster execution of input parallel strategies. Additionally, it can be seen in figure 5c, which includes an additional 1024 KiB configuration, that the cycle count does not scale linearly with the available memory. Instead, a combination of larger vector ALU with 128 processing elements (PEs) and 1024 KiB of memory can be faster than a configuration with 64 PEs and 256 MiB of memory. The reason is, that the 1024 KiB configuration only requires tiling of the initial layers of the network, which allows it to compensate for the additional cycles in the later layers where it can calculate twice as many results per cycle as the 64 PE configuration with 256 MiB of memory. A benefit of the simulator is the ability to quickly evaluate the performance for different networks across multiple configurations of memory sizes, vector ALUs and ISA variant.

## 6 Conclusion

This evaluation has shown that, while all evaluated toolchains deliver reasonable performance across different platforms, their support for low-powered embedded devices and heterogeneous solutions with dedicated accelerators is still at an early stage. They are limited by the use of existing compiler backends, which prevent the targeting of dedicated hardware. Another limitation is the use of static quantization schemes that do not offer an easy solution to adapt them for different hardware implementations.

Additionally, while it was possible to target a novel accelerator using TVM, our implementation showed two drawbacks: The more flexible quantization flow of TVM introduces annotation nodes into the code generation, that prevent the solution from reaching higher efficiency. It is currently not possible to connect the BYOC flow with the micro-TVM runtime, which is still under development. This prevents its use in conjunction with microcontrollers and Cortex-M CPUs.

An observation during the testing was that the coarse-grained ISAs of the accelerator is hard to adapt to new layer and operation types. While these kind of ISAs achieves good performance results by using extensive pipelines, they are inflexible. This can be a limitation for hardware solutions that are supposed to be sold for longer time periods, as they are at the risk to quickly become obsolete, if they cannot be adapted to new layer and function types efficiently. This is especially problematic due to the fast innovation speed of the deep learning research community.

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
