# OpenReview forum: "Compiler Toolchains for Deep Learning Workloads on Embedded Platforms"
_tinyml.org/tinyML/2021/Research_Symposium — tinyML 2021 Regular_

### Official Review · AnonReviewer2 · 2021-01-20

**Overall Merit Score:** 3

**Brief Summary:**

The paper provides a survey of several different tool chains and frameworks for mostly processor based design.  The paper compares and contrast different tool chains applied to different processor architectures.   The paper discuss some of the merits and different approaches these tool chain solves the problem and some of its consideration.

**Detailed Comments:**

The community would benefit from a good survey paper that compares different tool chains and its implication on memory, and inference time. From this, it may be easier to compare the different approaches in terms of run time and memory.

**Paper Strengths:**

- The paper is a good survey to compare, contrast, and highlight different tool chains.  It includes most of the popular flows and a couple of flows in the second tier.
- The paper some metrics of the different approaches.
- Figure 3 showing the different inference times of ARM A78 and J1900 is interesting in that one is not always ahead of the other and the margins are quite different.

**Paper Weaknesses:**

- Most of the targets used are outside the class of TinyML.  Some of these can be extrapolated with work.  The smallest processor is a M55.  A large portion of the data is from A72 and above.

**Poster (If Paper Is Rejected):**

1: Yes, ok for poster sesion to nurture work

**Reviewer Confidence:**

4: The reviewer is confident but not absolutely certain that the evaluation is correct

---

### Official Review · AnonReviewer3 · 2021-01-29

**Overall Merit Score:** 2

**Brief Summary:**

The first part of this paper is a survey and benchmark of the available open-source deep learning compiler toolchains targeting embedded microcontrollers combined with a dedicated accelerator.

The second part focuses on the implementation and evaluation of a compilation flow that uses one of the existing toolchains to demonstrate steps for hardware developers to build a software flow for their product.

**Detailed Comments:**

Below are some suggestions and comments which could help improve the paper.

Define explicitly what low power embedded systems are that the
toolchains of this paper target.

Aggregate Tables 1, 2, and 3 and analyze them together.

In Section 3.5, describe what was learned from the benchmarking.

The three charts in Figure 5 should have accompanying analysis. What should
the reader be taking away from them? Currently, there is no trend, and the
y-axis on each figure is different. An explanation is needed to
describe the "configurations." Also, requant cycles should be
defined.

The simulator used by the paper sounds important. Please provide
details about it. Can the simulator support ll ISAs (or
programming models) ranging from x86 to TPUs to GPUs?

**Paper Strengths:**

The paper provides a comprehensive survey on compiler toolchains
targeting embedded microcontrollers. It reports empirical
performance results of commonly used libraries by benchmarking
memory usage and inference time. It also describes an
implementation demonstrating the work required to compile to a
new platform.

Some of the paper's content could be useful to practitioners in
this field, especially to those engineers and students entering
into the field.


**Paper Weaknesses:**

The paper is not well-written.  Often it is hard to understand
what the authors are doing, and how and why they are doing it; the paper
reports empirical performance results without providing much
analysis, nor suggestions for future improvements.

At times, the paper is vague where it matters. For example, for the
software flow TVM discussion, the paper does not describe how
compute graphs from CNNs are partitioned into subgraphs, and how
assembly code is generated.

For the performance results of Figure 5, the paper does not provide
breakdowns of where most of the clock cycles are spent, so it is difficult to know what the results mean.


**Poster (If Paper Is Rejected):**

1: Yes, ok for poster sesion to nurture work

**Reviewer Confidence:**

4: The reviewer is confident but not absolutely certain that the evaluation is correct

---

### Official Review · AnonReviewer1 · 2021-01-30

**Overall Merit Score:** 3

**Brief Summary:**

This paper looks over existing approaches to compilation for deep learning on embedded systems, along with a case study examining how to deploy such an approach in practice.

**Detailed Comments:**

I can see a lot of hard work went into this paper, and it has very useful information for researchers and developers in this field.

**Paper Strengths:**

The strengths, with notes inline, are:

### Comprehensive overview of existing work

The summary of the different solutions currently available for embedded ML is detailed, knowledgeable, and well-organized to help highlight the strengths and weaknesses of each approach. The use of tables to cover the different capabilities is helpful, as is the discussion of the multiple different elements (like math libraries) that make up most systems.

### Example implementation

The account of the authors' experiences implementing a TVM workflow for a new accelerator was illuminating, as it uncovered challenges that might not have been visible from inspection of the tools. The observations around the problems of fixed-function accelerators in a world of rapidly-changing model architectures were useful too.

### Metrics

Having comparable data across many of the frameworks will allow potential users to make better educated choices between them, which is an important contribution to the field.

**Paper Weaknesses:**

### Limited experiments

I would have liked to see a wider variety of tests run to produce metrics. For example, figure #2 shows memory usage, but it appears the maximum is only 3KB. It would be helpful to also have data points for a larger model, to tell if there are fixed offsets or if overall memory usage is separated by scale factors between frameworks (for example does TFLM use 2x memory overall compared to others, or 1.5kb + 1x, or some other combination?). Similarly, the use of a network whose minimum latency time is less than a millisecond might hide constant-time factors that are amortized out for more computationally-intensive models, so more data points would help characterize performance there too.

## No information on op or model coverage

In practice a very important factor in choosing a compiler or framework is which models and operations it supports. There's information about file formats, but support for a file format doesn't mean that all, or even many, models produced in that format can be run. It would be helpful to see each approach presented with a representative selection of models to see how well they convert, if at all, to understand what coverage of common requirements they offer in practice.

**Poster (If Paper Is Rejected):**

1: Yes, ok for poster sesion to nurture work

**Reviewer Confidence:**

5: The reviewer is absolutely certain that the evaluation is correct and very familiar with the relevant literature

---

### Official Review · AnonReviewer4 · 2021-01-31

**Overall Merit Score:** 2

**Brief Summary:**

The paper presents a summary of the state of the field of deep learning compilers for embedded platforms.  The authors describe several compiler toolchains for running deep learning models on embedded hardware, and present performance and memory footprint measurements.  The authors also describe the process of porting one of the toolchains to a new accelerator.


**Detailed Comments:**

* It is not clear from the abstract whether the paper is a tutorial on existing workflows or a description of a new proposed workflow.  After reading further, it appears that the paper is primarily tutorial, but the target audience or the primary purpose of the paper is still not clear.  Is it an embedded engineer trying to decide which toolchain to use for a TinyML project?  Is it a hardware developer with a custom accelerator trying to decide with toolchains to support?

* Section 3.2 "The user accessibility mostly depends ..." -> "User accessibility mostly depends ..."

* Section 3.2 Mechanics: "ONNX[37] is the most important as it is an industry standard and converters from most frameworks exist." Add a comma after important, converters -> converts, "frameworks exist" -> "frameworks that exist".

* Figure 4: The Celeron is a large processor relative to what is typically considered "TinyML" and the memory allocations shown in this table are all quite large.  It is not clear how informative this table is for engineers exploring Tiny use cases.

* Section 3.5 Of the targets tested for performance, ARM Cortex-M55, Cortex-A72, and Intel Celeron J190016, only the M55 is applicable to TinyML use cases. Similarly, of the hardware listed in Table 3, Cortex-M is the only commercially available processor suitable for Tiny use cases.  "Bare Metal" is not hardware; it is a software architectue (or lack thereof).  NVidia's DLA is available as verilog, but to this reviewer's knowledge it is not available as a chip.



**Paper Strengths:**


* The paper presents a detailed description of several different toolchains.
* The performance and memory allocation data is detailed and should be useful.


**Paper Weaknesses:**

* The paper provides a great deal of factual information about different toolchains, but the implications for an engineer in any given situation are not always clear.  It is not very clear who should be reading this paper and what they should hope to take away.
* The fit to the conference scope of Tiny ML is marginal, because some of the platforms investigated (x86, CUDA GPU, Cortex A) are larger than most Tiny use cases can afford.



**Poster (If Paper Is Rejected):**

1: Yes, ok for poster sesion to nurture work

**Reviewer Confidence:**

4: The reviewer is confident but not absolutely certain that the evaluation is correct

---

### Decision · Program_Chairs · 2021-02-05

**Decision:**

Accept (Regular)

**Comment:**

Congratulations on your paper's acceptance!

Your paper has been accepted as a full-length regular paper.

Please read the reviews carefully and make sure the concerns are addressed in your final submission.

All accepted papers will be given a slot in the TinyML Summit schedule for an oral presentation on Friday, March 26, 2021.

Camera ready instructions will follow soon. All papers will be hosted on arXiv and published papers will have the following header stamp: “Published as a conference paper at TinyML Research Symposium 2021.” The paper will also be presented on the program website.